# Long-Term Yield Variability of Triticale (×*Triticosecale* Wittmack) Tested Using a CART Model

**Elżbieta Wójcik-Gront \* and Marcin Studnicki** 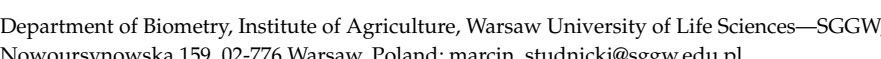

Department of Biometry, Institute of Agriculture, Warsaw University of Life Sciences—SGGW, Nowoursynowska 159, 02-776 Warsaw, Poland; marcin_studnicki@sggw.edu.pl
\* Correspondence: elzbieta_wojcik_gront@sggw.edu.pl

**Abstract:** Triticale is a promising food crop that combines the yield potential and grain quality of wheat with the disease and environmental tolerance of rye. The objective of this study was to evaluate the impact of genotype, environment and crop management on spring and winter triticale yield variability, using data from 31 locations across the whole of Poland, from 2009 to 2017, with the Classification and Regression Tree (CART) analysis. It was found that CART is able to detect differences in spring and winter triticale successful growth. The yield variability of spring triticale was more dependent on the soil quality than winter triticale because of a shorter cycle duration, which increases sensitivity to nutrient supply and weather conditions. Spring triticale also needs to be sown as soon as possible to ensure a successful establishment. A strong dependence of yield variability on the availability of water for the winter triticale was observed. When growing winter triticale in Poland, with periodic excess water especially during autumn and early spring, the use of fungicides and growth regulators should be taken into account.

**Keywords:** cereal yield; classification and regression tree; CART; sowing date; pesticides

## 1. Introduction

Triticale (×*Triticosecale* Wittmack) is a hybrid of wheat (*Triticum* ssp. used as the female parent) and rye (*Secale cereale* L. as the male parent) first bred in 1875 [1]. Its name is derived from the Latin terms for its parents, *Triticum* and *Secale*. Triticale was bred to combine the yield efficiency and grain quality of wheat with the disease and environmental endurance of rye [2–5]. It was reported that triticale can perform better than wheat on poor soil quality [2,6]. So far triticale is grown mostly as a feed grain, cover crop and for biogas production [5]. Although, triticale contains gluten, it may play a role in the rising healthy food market due to its health benefits with its good essential amino acid balance, minerals and vitamins [7,8]. Modern cultivars of triticale can be used for ethanol production [9,10]. Triticale combines good quality of grain with high levels of protein and lysine and is productive with low input requirements, is less susceptible to the common fungal diseases of cereals and has better adaptation to waterlogged soils, alkaline and acid soils, and nutrient deficient soils than other cereals [11–16]. However, winter and spring triticale grain yield production have not yet been assessed broadly across an array of environments, genotypes and managements in Poland [17–19]. Triticale world production has kept growing during the last two decades. According to the Food and Agriculture Organization, nowadays 15.5 million tons of triticale are harvested in 41 countries across the world [20]. The primary triticale producer in the world is Poland [20]. Following the statistics of Poland [21], total triticale production was over 5.3 million tons in 2017. That includes winter triticale (4.7 million tons) and spring triticale (0.6 million tons). In 2017, 86% of triticale acreage was covered by winter triticale, giving 89% of triticale yield. The small share of spring triticale in the triticale growing area (14%) is mainly due to its lower yield potential in comparison to winter triticale, because its cycle duration (i.e., the phase from sowing to the time when plats are harvested is shorter and it is heavily dependent

on weather and agricultural management). The spring cereals might supplement winter cereals production when disadvantageous autumn weather conditions, or frost in the wintertime, during the emergence occur causing winter cereals damage, resulting in yield loss [22]. Recently, in temperate climate environments it can be observed that winter plants are damaged by frost due to the lack of snow cover [23]. It is observed mainly for rapeseed but also for cereals. However, farmers are not convinced enough to use spring cereals to obtain better quality parameters of grain because the yield is lower. They use more frost-tolerant cultivars, if available, instead. When spring forms are used, spring triticale is a good alternative cereal forage crop to barley and oats [22]. When compared to spring wheat or barley, spring triticale showed superior yields on marginal lands and in drought conditions [12,22].

The physiology of spring and winter triticale is so different that they require different agricultural management and react differently to environment- and genotype-related variables. The problems of yielding spring forms are associated with disturbances in the field of water supply, soil pH and balance of nutrients in soil [24]. They are usually compounded by weeds, diseases and pests. Spring cereals have a shorter growing season and weaker root system than winter forms [25]. On the other hand, winter triticale needs fungicide protection due to long and wet fall and short cold summer seasons in moderate climate [19].

Triticale in Poland grows and is harvested similarly to most cereal crops, maturing in early to late summer. It is well known that cereal yields are usually driven by climate related variables that interact with the soil [17–19]. Despite high environmental influence, both agronomic and economic performance of triticale cultivation promote an agricultural intensification. Therefore, it makes sense to take the variability of triticale performance into account to develop better recommendations to increase crop productivity, profitability and fertilizer use efficiency [19]. Thus, the focus of this work is to inform about the main drivers of yield variability in spring and winter triticale cultivation. For this purpose, the Classification and Regression Tree (CART) was used as an appropriate tool for the analysis of yield variability. Furthermore, we wanted to check if using fungicides and growth regulators in winter triticale production is as important as in winter wheat production in Poland. That might suggest that in all winter cereals cultivated in Poland, a protection against fungus and weather-related problems like flooding during the harvest season is necessary. It was also interesting, if CART is able to detect growing conditions as the most important in spring triticale cultivation in Poland.

## 2. Materials and Methods

We assessed the impact of variables related to genotype, environment and crop management on the yield of 55 winter triticale genotypes tested across 61 locations and 13 spring triticale genotypes tested across 31 locations during 8 growing seasons (years 2009–2017). Not all cultivars were tested in each location and each year. That led to 12,352 observation units of winter triticale and 2020 for spring triticale. The study was performed using the CART analysis [26].

### 2.1. Experimental Data

Yield data of spring and winter triticale were gathered from the Polish Post-Registration Variety Testing System (PRVTS), where the yield and other related traits of newly released cultivars are evaluated in multi-environmental trials. The data used in this study contained observations from moderate input intensity with mineral fertilization including nitrogen, phosphorus and potassium adapted to the conditions in each location, the interventional use of herbicides and insecticides and seed treatment, and from high input intensity with additional (40 kg ha$^{-1}$ yr$^{-1}$) nitrogen fertilization, use of foliar fertilization, fungicides and growth regulators which were not applied in the moderate intensity system. Each field experiment was conducted according to a two-factor (agricultural management and cultivar) strip-plot design with two replications using a resolvable incomplete block design

for cultivar plots. The harvest area of each plot was 15 m². Winter triticale was sown during end of September and beginning of October and spring triticale was sown during end of March and beginning of April. Both crops were harvested during end of July and August. For each yield observation weather, soil, the length of the cycle duration, genetic and management related variables were described (in brackets—variable types in CART):

- The amount of seeds (quantitative);
- The rate of nitrogen, phosphorus, potassium and foliar fertilization (quantitative);
- The amount of fungicides, herbicides, insecticides and growth regulators per hectare (quantitative);
- Agro-region of Poland [19], from 1 to 6 (Figure 1) (qualitative);
- Climatic water balance (CWB), introduced by the Institute of Soil Science and Plant Cultivation (IUNG), is the basic indicator to estimate the water available for plants based on precipitation and potential evaporation [27]—in the analysis CWB was used for April and May, for May and June and for June and July during each growing season; variable was partitioned into 16 bins from sufficient water availability > −50 mm to extreme water deficiency −199 mm (quantitative);
- Cultivar (qualitative);
- Soil quality (valuation classes according to the soil quality evaluation system in Poland compatible with regulations of the Council of Ministers; class reflects the agricultural value of soils, the lower the class the more fertile the soils) [28,29] (quantitative);
- Date of sowing (quantitative);
- The number of days from sowing to harvesting (quantitative);
- Pre-crop (cereal, legume, rapeseed, root crop) (qualitative).

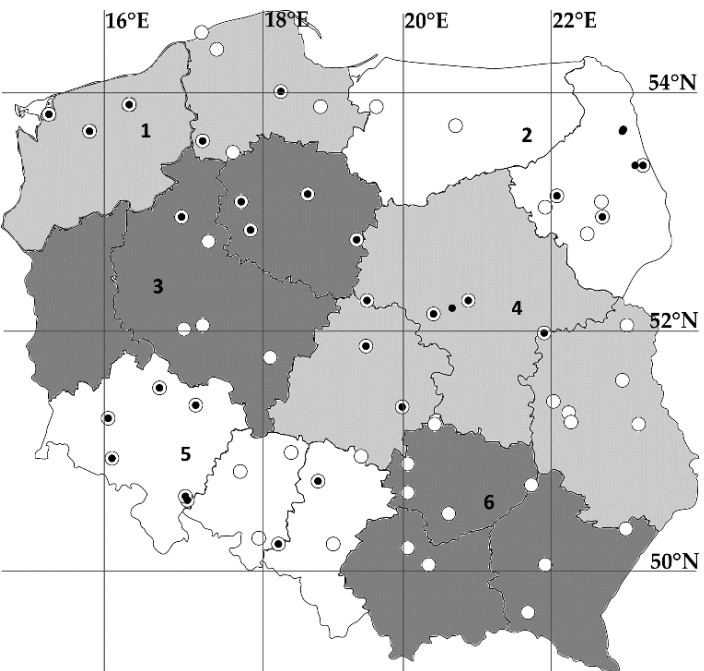

**Figure 1.** Locations of the winter triticale (blank dots) and spring triticale (solid dots) experiments which took place between 2009/2010 and 2016/2017 in Poland. Lines display the 16 Polish administrative units (voivodeships) and shades indicate the six regions used in the analysis. Grid straight lines represent the geographic coordinate system for Poland.

## 2.2. Statistical Analysis

The analysis of the main drivers of both winter and spring triticale yield variation was performed using the CART model [26] created with STATISTICA software ver. 13 [30]. CART is a non-parametric data analysis which predicts the value of a dependent variable by

the recursive division of data into smaller and smaller groups in order to create subsets with dependent variable as homogeneous as possible [31]. Here, the dependent variable was triticale yield and independent variables were the treatments and parameters investigated within the field trials. Initially, the whole data was divided into two data subsets. The optimal split was searched based on all splits for each independent variable to minimize dependent variable variation in created subsets. The process was then repeated for each subset until a subset can no longer be divided. In case of this work, the CART method was used to solve regression problems where the dependent variable is a continuous type variable. CART is a flexible statistical method appropriate for this study. This method has a number of advantages over alternative methods, such as logistic regression. CART needs a considerable number of observation units but because it does not require statistical distribution assumption regarding used variables there is no need for data transformation. It works well with unbalanced (like in this work) and missing data, and quantitative, qualitative data and their combinations. The single CART model can overreact to random variation in the data lowering its predictive performance, so we used a 10-fold cross validation method to "prune" overgrowth trees. CART also produces the independent variable importance ranking in terms of their impact on the dependent variable thus, their contribution to the constructed tree (i.e., splitting power calculated from the highest for the top variable, importance 1, to the smallest for the variable that is never featured in the tree, importance 0). CART description and use is well documented in literature [19]. Dacko et al. [32] used CART for determining biological and environmental factors influencing mass of pea. Krupnik et al. applied this method for untangling crop management and environmental influences on wheat yield variability in Bangladesh [33]. Mézière et al. used it to determine the best cropping system to reconcile weed-related biodiversity and crop production in arable crops in France [34]. Aman and Bhatti predicted potential yield of wheat in Pakistan depending on soil properties using CART model [35]. Aouadia et al. analyzed, with the use of this method, the impact of the farming context and environmental factors on cropping systems in Burgundy [36]. Work by Andrianasolo et al. presents the use of CART to predict sunflower grain oil concentration as a function of cultivar, crop management and environment [37].

The input data for CART were prepared to not include outliers for more flexibility in translation to other studies. An observation was considered as an outlier if the value was greater than the 75th percentile + 1.5·(the 75th percentile–the 25th percentile) or lower than the 25th percentile −1.5·(the 75th percentile–the 25th percentile). This led to 12,352 observations of winter triticale and 2020 spring triticale used for CART analysis.

In our trees, the stop rule was trimming to variance, the minimum number of observations in a shared node was 10% of the total number of observations (in case of winter triticale it was 1235 and for spring triticale 202), validation of the model was done with the 10-fold cross-validation with the rule of standard error equal 1.

Based on [32], we assumed the following criteria to estimate if the independent variable had an important impact on the dependent variable (importance: 0.0–0.3: Not important; 0.3–0.6: Moderately important; 0.6–0.8: Important; and 0.8–1.0: Very important).

## 3. Results

In Table 1 are shown medians, the first and third quartiles, and minimum and maximum values for quantitative variables related to the management of winter and spring triticale. In many cases no fertilization nor plant protection were used. Thus, for many observations the minimum values of these treatments were zero.

Additionally, the yield of winter triticale was higher than in the case of spring triticale (8285 vs. 6860 kg ha$^{-1}$).

### 3.1. Results for Winter Triticale

The CART analysis of triticale yield from eight growing seasons revealed differences in the importance of variables influencing yield between spring and winter triticale. The

variable giving the most yield variability reduction is growth regulator (Figure 2). When applied in doses smaller than 0.08 kg ha$^{-1}$ (which in case of our data is 0) it enables a yield of 7607 kg ha$^{-1}$. Higher (non-zero) doses of growth regulator result in yield 9111 kg ha$^{-1}$. For no doses of growth regulator next variable in CART is CWB for May and June related to seasonal weather. Lower yield is obtained for high precipitation in May and June. The next variable in this branch of the tree is soil class. In general, higher yields are obtained on soils of better quality. However, in drier conditions, on lower quality soils (IVa, IV, V, VI), high triticale yield can be obtained using doses of herbicides higher than 3.5 kg ha$^{-1}$, even with no doses of growth regulators (Figure 2 split ID 25). In case of no doses of growth regulators in drier conditions, on better quality soils (I, II, IIIa, IIIb), high triticale yield can be obtained with low doses of herbicides ($\leq$0.08 kg ha$^{-1}$) (Figure 2 split ID 16). The same was observed for wetter conditions (Figure 2 split ID 10). Still, in general it is crucial to use fungicides and growth regulators in winter triticale production. In our study these chemicals are used in the higher level of management intensity (correlation r = 0.72). Then, in regions 1, 2 and 3, high yield is obtained in drier weather in June and July with doses of growth regulators higher than 0.08 kg ha$^{-1}$. It was shown that, under zero dosages of growth regulators and fungicides, soil quality becomes an important variable determining the yield. On better quality soils, lower doses of herbicides give higher yield in contradiction to lower quality soils when doses of 3.5 kg ha$^{-1}$ are recommended. The coefficient of Pearson correlation between the observed and predicted dependent variable yields was 0.58. Variables which were less important in explaining winter triticale yield variation are: $K_2O$, date of sowing, insecticides, number of days from sowing to harvesting, pre-crop, $P_2O_5$, site conditions and foliar fertilization (Figure 3).

**Table 1.** Medians, the first and third quartiles and minimum and maximum values for quantitative variables related to triticale management.

| | Winter Triticale (kg ha$^{-1}$) | | | | | Spring Triticale (kg ha$^{-1}$) | | | | |
|---|---|---|---|---|---|---|---|---|---|---|
| | **Median** | **Q1** | **Q3** | **max** | **min** | **Median** | **Q1** | **Q3** | **max** | **min** |
| yield | 8284.4 | 6888.0 | 9760.0 | 14,063.7 | 2486.7 | 6857.8 | 5839.9 | 7900.2 | 11,037.6 | 2722.4 |
| seeds | 162.0 | 141.8 | 162.0 | 202.5 | 121.5 | 182.3 | 182.3 | 202.5 | 243.0 | 121.5 |
| N | 120.0 | 98.0 | 140.0 | 214.0 | 45.0 | 90.0 | 70.0 | 110.0 | 186.5 | 40 |
| $P_2O_2$ | 50.0 | 40.0 | 60.0 | 90.0 | 0 | 60.0 | 38.0 | 60.0 | 109.0 | 0 |
| $K_2O$ | 90.0 | 70.0 | 90.0 | 150.0 | 0 | 90.0 | 75.0 | 96.0 | 161.0 | 24 |
| foliar fertilizer | 0 | 0 | 3.2 | 80.0 | 0 | 0 | 0 | 1.6 | 24.7 | 0 |
| herbicides | 1.1 | 0.5 | 1.4 | 5.2 | 0 | 0.8 | 0.3 | 1.2 | 2.8 | 0 |
| insecticides | 0.1 | 0 | 0.2 | 1.6 | 0 | 0.1 | 0.1 | 0.2 | 1.2 | 0 |
| fungicides | 0 | 0 | 1.6 | 4.0 | 0 | 0 | 0 | 1.3 | 3.2 | 0 |
| growth regulators | 0 | 0 | 0.8 | 1.9 | 0 | 0 | 0 | 0 | 0.6 | 0 |

### 3.2. Results for Spring Triticale

In contrast to winter triticale, the sowing date is one of the main spring triticale yield variability influent variables. Early sowing extends the growing season of spring triticale which compared to winter triticale is much shorter (average 132 vs. 312). Whereas the harvest of both is around the same time. In the spring triticale tree the variable giving the most variability reduction in yield was CWB June-July (Figure 4). When values of this variable were higher than −50 mm, the yield was 5657 kg ha$^{-1}$, which is almost 1500 kg lower than for drier conditions (Figure 4 split ID 1). Yields higher than 8000 kg ha$^{-1}$ was obtained mostly for drier conditions in June-July, during the harvest season, and for longer cycle duration (earlier sowing date), and obviously on better quality soils. Coefficient of Pearson correlation between observed and predicted dependent variable yield was 0.68. Variables which were on the less important site in explaining spring triticale yield variation were: growth regulator (which was hardly ever used), foliar fertilization, cultivar, herbicides and seeds (Figure 5).

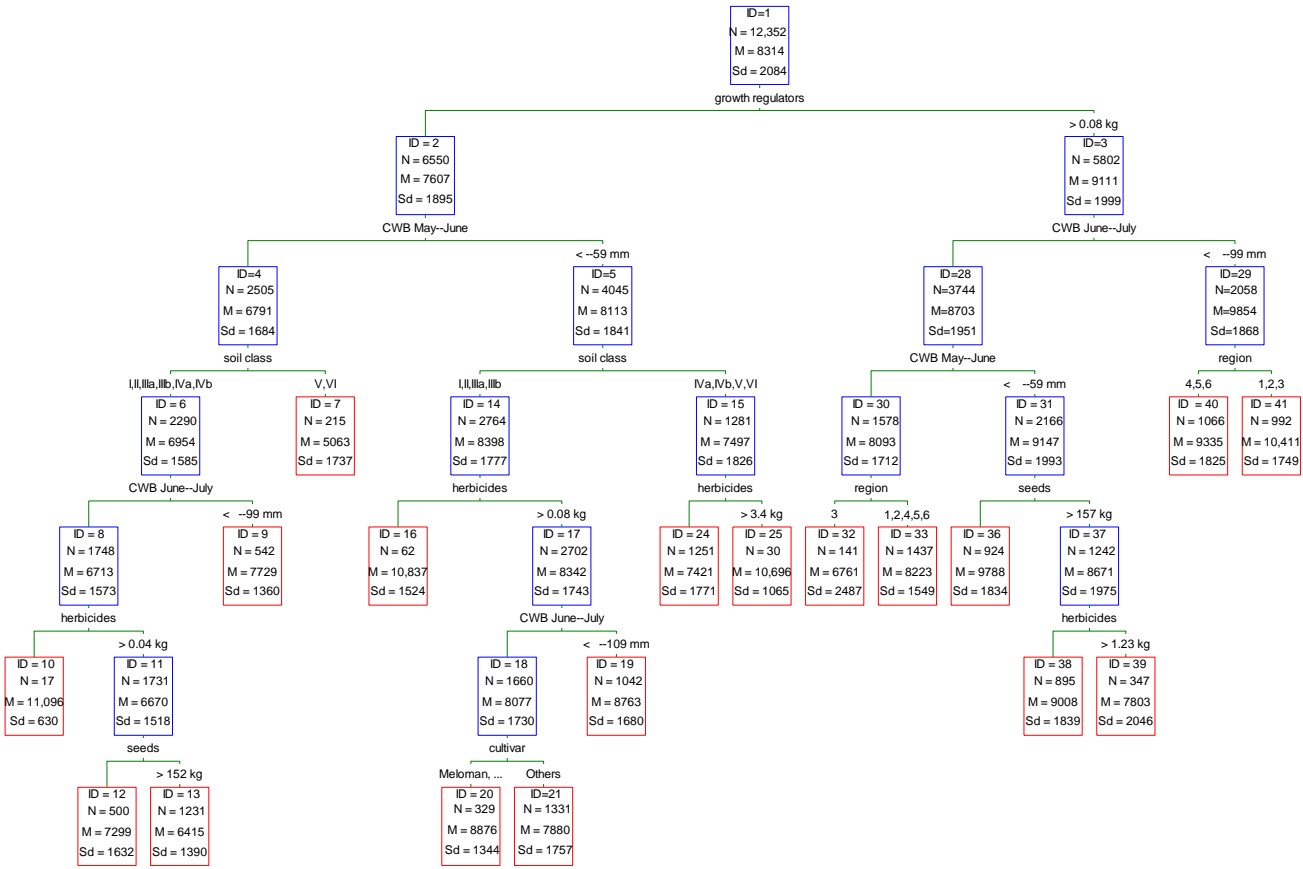

**Figure 2.** CART (Classification and regression tree) for winter triticale. CWB stands for Climatic Water Balance. N indicates the number of observation units in each subset. There are also mean (M) and standard deviation (Sd) in each subset given. In split ID, 20 cultivars giving higher yields are as follows Meloman, Subito, Panteon, Trapero, Pizarro, Avokado, Festino, Kasyno, Rufus, Sekret, Temuco, Sorento, Witon, Trisol.

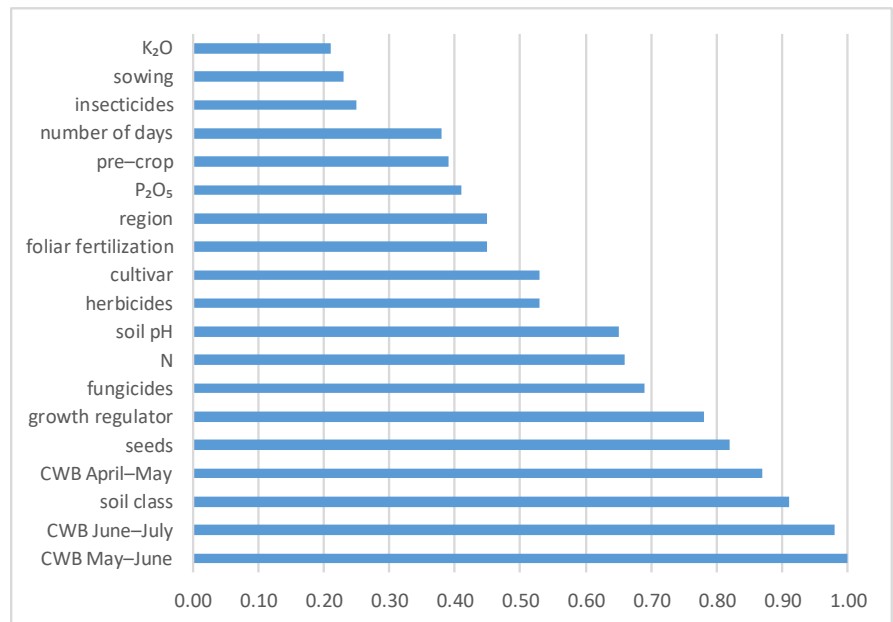

**Figure 3.** Importance of independent variables in the scale 0–1 (importance 0—variable is never featured in the tree, importance 1—the highest importance in the explanation of yield variability) for winter triticale.

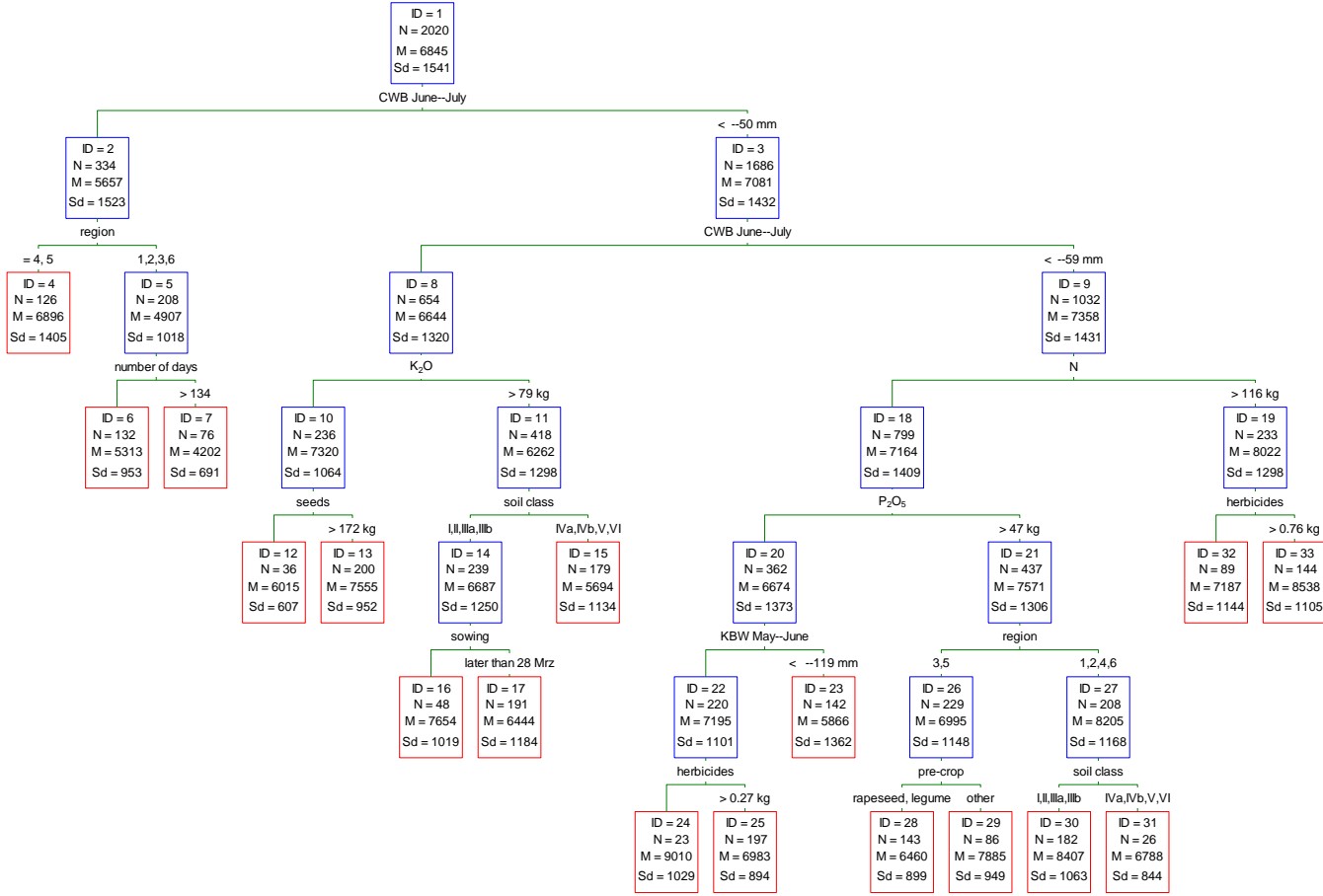

**Figure 4.** CART (Classification and regression tree) for spring triticale. CWB stands for Climatic Water Balance. N indicates the number of observation units in each subset. There are also mean and standard deviation (Sd) in each subset given.

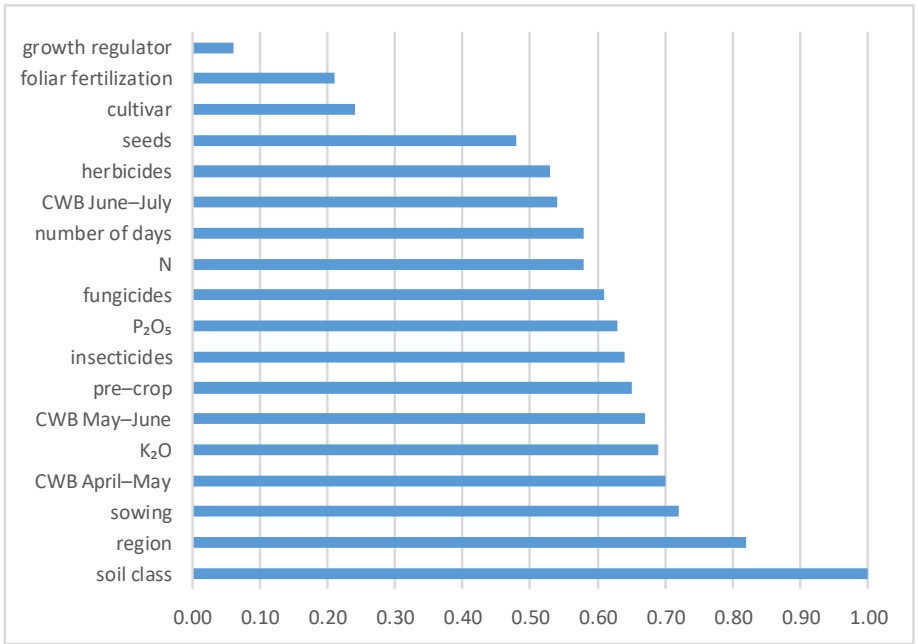

**Figure 5.** Importance of independent variables in the scale 0–1 (importance 0—variable is never featured in the tree, importance 1—the highest importance in the explanation of yield variability) for spring triticale.

## 4. Discussion

In general, CART detected significant differences in agricultural management of spring and winter triticale. In case of spring triticale, the sowing date had one of the greatest impacts on yield variability. The same was determined by Cheshkova et al. [38] using a mixed model to study genotype, sowing rate and sowing date effect on variation in spring triticale yield. Spring triticale requires the date of sowing as soon as possible to extend its cycle duration [39]. The optimal sowing date depends to a large extent on the weather and environmental variables—when they are unfavorable, the sowing date should be earlier to improve the probability of plant survivals and a successful establishment of the plants. Sowing should take place immediately after the soil is dry, if the farmer can enter the field. The optimal date for sowing spring triticale depends on the region of Poland and falls on the third decade of March or the first decade of April. In general, the western part of Poland is characterized by warmer climatic conditions, and the eastern part has lower average temperatures and shorter growing season [40]. Thus, the variable agro-region was very important in explaining yield variability in case of spring triticale and moderately important for winter triticale. The trial locations were chosen to be representative of the climatic and soil conditions of a given region. However, in case of winter wheat the agro-region was very important in explaining yield variability [19] and also a good agreement in yield between the trial locations was obtained [41]. In case of winter triticale, it might be necessary to increase the number of locations in the regions for future studies.

In general, starting from May, both spring and winter triticale prefers drier conditions with higher solar radiation to obtain higher yield according to the results presented here and other triticale studies [12]. A yield higher than 10,000 kg ha$^{-1}$ was obtained for several environmental- and management-related variable combinations. Unfortunately, sometimes the relations between variables is not monotonic and quite difficult to interpret. However, better soils do not require high input to give satisfying yield [19]. The variable soil class was the most important in explaining yield variability of spring triticale, which is more sensitive to nutrient availability and environmental conditions during its growing season. On the other hand, on lower quality soils high yield can be obtained by proper fertilizers and pests management [42]. Poor soils are characterized by low retention capacity of water and reduced nutrient availability, that might intensify the crop-weed competition and ultimately reduce crop yield [43,44]. Then herbicides might improve the available nitrogen, phosphorus and potassium level in soil by reducing nutrient mining by weeds. Sometimes the satisfactory weed control may be obtained with lower doses of herbicides which might be related to different composition of weeds on different quality of soils [45]. The results of this work showed that in the triticale production the disadvantages of lower quality soils can be partly compensated by using proper management which stays in agreement with other studies on cereals [46]. In case of this analysis, if N fertilization increases above 100 kg ha$^{-1}$ the yield is comparable to the one on better quality soils for both spring and winter triticale. Similar value of nitrogen application in triticale production was obtained by [47,48]. However, if N fertilizer is applied at a rate greater than can be taken by plants, with attempts to increase yields only by increasing inputs, this can contribute to agricultural inefficiency [19] and nitrogen leaching [49–51]. Urruty et al. [47] found that there is noticeable impact of agricultural practices on yield robustness to unfavorable weather conditions. In case of wheat grown in average weather conditions, yield was significantly higher when more nitrogen and additional chemicals were used [19]. This was found for the use of both fungicides and growth regulators in winter triticale. These chemicals can prevent yield loss due to fungal diseases resulting from water logging [12,13,52]. In triticale production there are also lower rates of fertilizers applied (33 kg ha$^{-1}$ N) [53]. Roques et al. stated that current N fertilizer recommendations for triticale in the UK are too low and triticale out-performs wheat on a range of UK soils with a similar nitrogen requirement [45]. According to Dumbravă et al., triticale, in comparison to wheat or maize, gives higher yield when intensive treatment is applied in the low quality soil or unfavorable weather conditions [22]. In general, triticale has the capacity to survive in soils which

would be considered nutrient deficient for any other type of crop [13,54]. Growth and yield of triticale is very responsive to phosphorus and nitrogen [55], which also can be seen in this study. In addition, relations between yield and soil physical properties change between years, mainly depending on weather conditions [56]. In the case of the presented study, yield dependence on weather is expressed by strong influence of CWB. It can have many implications due to the expected climate change effects on agriculture in Poland [57]. Higher average temperatures will extend the growing season in Poland. On the other hand, the warmer climate affects the occurrence of extreme weather events, such as droughts, floods and attracts new crop-destroying pests to Poland.

It turned out that the cultivars used were not important in explaining yield variability, neither in winter nor in spring triticale, even with so many levels of independent variable. In case of winter triticale, the variable cultivar has 55 levels (winter triticale cultivars) and 13 levels for spring triticale. When an independent variable has many levels, CART can be biased toward choosing this variable [58]. However, in this work, cultivar was moderately important in explaining yield variability in the tree for winter triticale and not important in the spring triticale tree. This effect might be artificial due to many levels of this variable and in future might be compared with outcomes from unbiased recursive partitioning [59], proving that a cultivar is indeed not important in explaining triticale yield variability. In cultivars evaluation based on multi-environmental trials, modern, high-yielding cultivars are used. The effect of cultivars is not significant. In general, the genetic factor explains a very small percentage of yield variability (1–2%). This is particularly true for modern cultivars which generally provide a balanced yield [19,60].

This work shows that, among all the pesticides used, fungicides were the most important in explaining triticale yield variability [19], especially in case of winter triticale, although triticale can be less susceptible to the common fungal diseases of cereals. However, its healthiness has been steadily declining with the expansion of triticale area [61]. One point for discussion may be the pesticide dose. Pesticides from various companies were used in the experiments. Our assessment shows that the active ingredient content is similar for each type of chemical. Therefore, we decided to include pesticides and growth regulator as continuous variables in the analysis. Additionally, we performed a CART analysis with the assumption that these chemicals would be included as a zero–one variable, that is, 0—no use, 1—used without specifying its dose. The results turned out to be very similar to the analysis presented here. Which proves that, in the case of winter triticale and other winter cereals grown in Poland, it is important to use a certain dose of fungicides and growth regulators. Overall, triticale requires a similar pest control as wheat [19] and probably other winter cereals grown in Poland. Non-monotonical dependence on pesticide doses can be seen in the results. The solution would be the integrated production of [62], which is plant cultivation to maximize yields while minimizing costs resulting from the limited use of chemical.

## 5. Conclusions

In the presented study it was found that CART is able to detect variables influencing the effective growth of spring and winter triticale. The yield variability of spring triticale was more dependent on environmental conditions during its cycle duration. The variable in the production of spring triticale which is influenced by the farmer is the sowing date, it should take place as soon as possible. As in the case of other winter cereals cultivated in Poland, a strong dependence of the variability of the yield on the availability of water for winter triticale was found. In the cultivation of winter triticale in Poland, with periodic excess of water, especially in autumn and during the harvest, the use of fungicides and growth regulators has to be included.

**Author Contributions:** Conceptualization, methodology, formal analysis, investigation, resources, data curation, writing—original draft preparation E.W.-G., writing—review and editing M.S. All authors have read and agreed to the published version of the manuscript.

**Funding:** This research received no external funding.

**Conflicts of Interest:** The authors declare no conflict of interest.

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
