# Peer review of "Long-Term Yield Variability of Triticale (×Triticosecale Wittmack) Tested Using a CART Model"

_agriculture, doi:10.3390/agriculture11020092_

Round 1

Reviewer 1 Report

The paper appears well written and clearly structured.
The topic is interesting both from a methodological point of view and from that of the results concerning the cultivation of triticale.
In the description of materials and methods, however, it would be appropriate to deepen some details.

My observations are listed below in order of importance:

The definitions of the independent variables "CWB" and "soil quality" (lines 111-117) refer to articles in Polish and are therefore not accessible to foreign readers. Since these variables are of pivotal importance in the work, it would be better to report in the text more detail on how the calculation of these indices is carried out.

The input of insecticidal herbicides and growth regulators (line 107) is defined in kilograms regardless of the type and dose of use. A discussion of the types of active substances included in the individual classes and a discussion of whether or not weights should be normalized according to dose of use would be appropriate.

In fgure 4 and 5 the use of standard deviation instead of variance (V) would make the decision tree graphs more readable.

The graphical rendering of Figure 1 could be improved.

The statements shown on lines 50-50 would need reference.

Author Response

In the description of materials and methods, however, it would be appropriate to deepen some details.

Thank you for the valuable comments.

The definitions of the independent variables "CWB" and "soil quality" (lines 111-117) refer to articles in Polish and are therefore not accessible to foreign readers. Since these variables are of pivotal importance in the work, it would be better to report in the text more detail on how the calculation of these indices is carried out.

We changed or added the references to articles in English which in details discuss the indices used in our work.

Szewczak, K.; Łoś, H.; Pudełko, R.; Doroszewski, A.; Gluba, Ł.; Łukowski, M.; Rafalska-Przysucha, A.; Słomiński, J.; Usowicz, B. Agricultural Drought Monitoring by MODIS Potential Evapotranspiration Remote Sensing Data Application. Remote Sens. 2020, 12, 3411

Kabała, C., et al., 2019. Polish Soil Classification, 6th edition – principles, classification  scheme  and  correlations.  Soil  Science  Annual  70(2), 71–97.

The input of insecticidal herbicides and growth regulators (line 107) is defined in kilograms regardless of the type and dose of use. A discussion of the types of active substances included in the individual classes and a discussion of whether or not weights should be normalized according to dose of use would be appropriate.

The amount of insecticidal herbicides and growth regulators was determined in kg of dose before dilution per ha. Indeed, there were several types of preparations (e.g. Cerone, Moddus, and sometimes there was only information about the dose of chemicals without their name). Since the doses of the active substance in these preparations are usually similar, we decided to include these preparations as continuous variables. Our other idea was to include these plant protection products as a zero-one variable, i.e. 0 - no use, 1 - use without any dose information. We actually performed an analysis, where the amount of foliar fertilizers, herbicides, insecticides, fungicides and growth regulators were 0 or 1 and the tree structure or the importance of predictive variables did not change much. In general it is crucial to use fungicides and growth regulators in winter triticale production. There is a significant yield improvement when any doses of these chemicals are used. We included some discussion on this topic in the manuscript L331-339.

In figure 4 and 5 the use of standard deviation instead of variance (V) would make the decision tree graphs more readable.

The parameter was changed to standard deviation

The graphical rendering of Figure 1 could be improved.

We decided to remove the graph and present the data in a table. We hope that the data presented in this way is clearer.

The statements shown on lines 50-50 would need reference.

We provided reference: Kreyling, J. Winter Climate Change: a Critical Factor for Temperate Vegetation Performance. Ecology 2010, 91(7), 1939–1948.

Reviewer 2 Report

Dear authors,

I have reviewed your paper and I found it interesting. The topic is relevant for this journal.

The paper analyzes long-term data on winter and spring triticale cultivated into experimental trials in several locations in Poland. Yield is modeled on several agronomic and pedoclimatic factors, using nonparametric classification and regression trees.

The paper is well organized and well written, the objectives are clearly defined, and the methodology is clear and well described. The discussion section offers some interesting insights about the factors influencing triticale production in Poland, and is widely supported by bibliographic resources.

Overall, I have some minor observations and doubts that I hope the authors will clarify.

MAIN OBSERVATION:

My main doubt is about the correlation between the input variables used by the model to predict yield. There are some instances where I got the feeling that some variables selected by the tree could have masked some other variables that were excluded by it, or variables that were not considered in the study. For example, in line 248-250 drier conditions (with higher solar radiation) are pointed out to improve yield. Is it because of drier conditions, or because of greater radiation? Could the dryness-wetness of the period “mask” the amount of cloudiness? Another instance is in line 292-300, where the variable cultivar is said to be of minor or no importance, despite the high number of levels that should lead the model to prefer it in the choice. On the other hand, the variable agroregion had an importance rate of around 0.45 for winter triticale and of 0.80 for spring triticale. Could this variable mask the results from cultivars? Were the same cultivars tested in all the experimental fields or only within regions into which they are most adapted? Could the model have chosen the agroregion variable instead of the cultivar variable to predict yield, while yield was the result of using the most adapted cultivars within each agroregion?

In general, does the CART method consider possible correlations between independent input variables when building the tree?

OTHER MINOR OBSERVATIONS:

  • Materials and methods. Is there any “parameter” that needs to be set by the user when running the model (e.g. minimum number of observations or something related to pruning)? Or does the cross-validation take care of everything to get the best fit?
  • Line 87, 124, 139. Usually the term “multivariate” is used when there are multiple dependent variables, while in this study yield is the only output variable. In this case, is the term “multivariate” used to refer to multiple input variables, or were winter triticale yield and spring triticale yield considered at the same time when building the model?
  • Line 166-172. Should be Figure 2 (line 170). Please also improve the representation of your raw data. In some cases, it is difficult to interpret the min-max range given the axis labels (e.g. in “insecticides” both the left and right label is 0.08, so the scale is not clear). Please consider redrawing the graph or providing more textual support to it by expanding the description.
  • Line 194-195 and 216-217: For both models, please provide some more statistics about the model performance. E.g. graphs with observed vs. predicted are usually a good visual support for the readers to interpret residual variability and over/underprediction of the model (if appropriate for CART).
  • Please consider putting Fig. 3 and Fig. 5 full page with landscape orientation if allowed by the journal guidelines. Right now it is difficult to read the numbers in the boxes.
  • Line 207. The section should be 3.2.
  • Line 241. From this line onward, expressions like “important”, “moderately important” or “very important” are often used. While the importance rate is a quantity, these expressions refer to something more subjective. Please define reference thresholds of “importance” in the M&M section (e.g. 0-0.2: not important, 0.2-0.4: moderately important, and so on). Is there any bibliographic reference for that?

Kind regards

Author Response

Overall, I have some minor observations and doubts that I hope the authors will clarify.

Thank you for the valuable comments.

My main doubt is about the correlation between the input variables used by the model to predict yield. There are some instances where I got the feeling that some variables selected by the tree could have masked some other variables that were excluded by it, or variables that were not considered in the study. For example, in line 248-250 drier conditions (with higher solar radiation) are pointed out to improve yield. Is it because of drier conditions, or because of greater radiation? Could the dryness-wetness of the period “mask” the amount of cloudiness? Another instance is in line 292-300, where the variable cultivar is said to be of minor or no importance, despite the high number of levels that should lead the model to prefer it in the choice. On the other hand, the variable agroregion had an importance rate of around 0.45 for winter triticale and of 0.80 for spring triticale. Could this variable mask the results from cultivars? Were the same cultivars tested in all the experimental fields or only within regions into which they are most adapted? Could the model have chosen the agroregion variable instead of the cultivar variable to predict yield, while yield was the result of using the most adapted cultivars within each agroregion?

In general, does the CART method consider possible correlations between independent input variables when building the tree?

Yes, some variables are indeed correlated. However, this issue is normally taken care of by the use of cross validation in the model. If two independent variables are correlated they would influence the tree in a similar way. When selecting splits, trees track the competitive splits at each decision point along the way. A competitive split is one that results in a similarly pure (in terms of variance) subset as the chosen split. When constructing the tree with a support of cross validation sometimes one variable would be chosen and sometimes the other correlated one. Thus, in CART both variables should have similar importance. Of course, we are limited to variables which were available at the time of data gathering. The variable CWB takes into account the state of humidification of the environment (assessment of current water resources) using meteorological data measured on ground metrological stations. While precipitation measurements are well established, the estimation of potential evaporation is more difficult and is based on solar radiation flux measurements. Regarding the agro-regions and cultivars, Poland has quite unified cultivar recommendation for the whole country area. Agro-region is a variable which introduces some climate differences in different parts of Poland. It means that indeed environmental conditions have significant influence on the yield, while modern cultivars give comparable yield.

Materials and methods. Is there any “parameter” that needs to be set by the user when running the model (e.g. minimum number of observations or something related to pruning)? Or does the cross-validation take care of everything to get the best fit?

Yes, we set couple of parameters. The materials and methods section was supplemented by additional information about the model input set-up.

“In our trees the stop rule was trimming to variance, the minimum number of observations in a shared node was 10% of the total number of observations, validation of the model was done with the 10-fold cross-validation with the rule of standard error equal 1.”

Line 87, 124, 139. Usually the term “multivariate” is used when there are multiple dependent variables, while in this study yield is the only output variable. In this case, is the term “multivariate” used to refer to multiple input variables, or were winter triticale yield and spring triticale yield considered at the same time when building the model?

The name was removed from the CART description. We created separate trees for winter triticale and for spring triticale.

Line 166-172. Should be Figure 2 (line 170). Please also improve the representation of your raw data. In some cases, it is difficult to interpret the min-max range given the axis labels (e.g. in “insecticides” both the left and right label is 0.08, so the scale is not clear). Please consider redrawing the graph or providing more textual support to it by expanding the description.

We have decided to present the data in the form of table. We hope that this presentation gives a better interpretation of the data.

Line 194-195 and 216-217: For both models, please provide some more statistics about the model performance. E.g. graphs with observed vs. predicted are usually a good visual support for the readers to interpret residual variability and over/underprediction of the model (if appropriate for CART).

The method divides the data into subsets. Each subset is defined by set of rules put on independent variables. The cross-validation method allows us to think of the presented tree as the best possible one. The only estimation of the tree performance is the calculated coefficient of Pearson correlation between observed and predicted dependent variable yield. In case of our analysis it was quite high considering that many observations with large variability. When comparing to other methods, this method has many advantages. It works well with unbalanced (like in this work) and missing data, and quantitative, qualitative data and their combinations. It would be difficult to find a regression model which allows to include all the independent, different in types variables at once.

Please consider putting Fig. 3 and Fig. 5 full page with landscape orientation if allowed by the journal guidelines. Right now it is difficult to read the numbers in the boxes.

Normally, the MDPI journals provide full resolution figures on their website. We will discuss that with the Editors.

Line 207. The section should be 3.2.

Yes, it was corrected.

Line 241. From this line onward, expressions like “important”, “moderately important” or “very important” are often used. While the importance rate is a quantity, these expressions refer to something more subjective. Please define reference thresholds of “importance” in the M&M section (e.g. 0-0.2: not important, 0.2-0.4: moderately important, and so on). Is there any bibliographic reference for that?

We included the description of the thresholds of “importance”: 0-0.3: not important, 0.3-0.6: moderately important, 0.6-0.8 important, 0.8-1.0 very important) based on:

Dacko, M., Zając, T., Synowiec, A., Oleksy, A., Klimek-Kopyra, A., Kulig, B., 2016. New approach to determine bio-logical and environmental factors influencing mass of a single pea (Pisum sativum L.) seed in Silesia region in Poland using a CART model. Eur. J. Agron. 74, 29-37.